# Language-guided Vision Model for Pan-cancer Segmentation in Abdominal CT Scans

Hui Meng[1][0000−0003−4061−2100], Deqian Yang[1][0009−0008−6633−3541], Nianjiang Lv[1][0009−0009−2764−1389], Haochen Zhao[2][0009−0007−9234−3594], and Yong Zhang[1][0009−0005−0174−9173]

[1] School of Intelligent Science and Technology, Hangzhou Institute for Advanced Study, University of Chinese Academy of Sciences, Hangzhou 310024, China
[2] Hangzhou Innovation Institute of Beihang University, Hangzhou 310051, China
`huimeng@ucas.ac.cn`

**Abstract.** Accurate segmentation of abdominal tumors is critical for clinical diagnosis, disease research, and treatment planning. Since deep learning-based segmentation techniques typically require a large amount of labeled data for training, it is crucial to develop precise segmentation methods that rely on smaller labeled datasets in medical image analysis. Recently, the advent of pre-trained vision-language foundation models, such as CLIP, has opened new possibilities for general computer vision tasks. Leveraging the generalization capabilities of these pre-trained models in downstream tasks, like segmentation, can achieve remarkable performance with relatively limited labeled data. However, the exploration of these models for tumor segmentation remains limited. Hence, in this paper, we propose a novel framework called the Language-guided Vision Model. Our approach employs the pre-trained CLIP as a powerful feature extractor to generate segmentations of 3D CT scans while adaptively aggregating cross-modal representations of text and images. On validation of FLARE 2024 challenge, our method achieves mean DSC of 43% and mean NSD of 38% on validation leaderboard for tumor segmentation.

**Keywords:** CLIP · VLM · CT.

## 1 Introduction

The abdomen is a common site for tumor growth. Accurate annotation of tumors in CT scans is crucial for the diagnosis and treatment of abdominal tumors. Despite the ease that deep-learning-based methods provide in the task of manual annotation for radiologists, several challenges still hinder their effectiveness. Segmentation methods for tumors have primarily focused on Convolutional Neural Networks (CNNs), particularly the U-Net architecture and its variants. These approaches have effectively leveraged the potential of limited labeled data, especially from CT scans. Many semi-supervised and weakly supervised learning approaches [29–31, 44] are proposed based on pseudo-labeling of the partially

labeled data. However, they often suffer significantly from the inaccuracy of pseudo-labels associated with unlabeled parts of the CT data.

Recently, the Vision-Language Model (VLM) pre-training and zero-shot prediction paradigm has garnered considerable attention. This approach entails pre-training a vision-language model using vast quantities of image-text pairs sourced from online platforms. Once pre-trained, such models can be directly utilized for downstream visual recognition tasks without the need for fine-tuning. For instance, CLIP [33] utilizes an image-text contrastive objective to align paired images and texts in the embedding space while disambiguating unpaired elements. Various attempts are underway to tailor VLMs for specific task domains. Some approaches [11, 34, 42] modify contrastive objectives to generative or alignment objectives for retraining a VLM. Conversely, other methods fine-tune existing VLMs at a lower cost, including techniques such as prompt tuning [20] and feature adapters [8]. In the context of medical image segmentation, models such as Self-Attention Module (SAM) [18] and its variants [23, 37] have been retrained on medical images (e.g., Med-SAM [23], SAM-MED3D [37]), achieving significant success in this area. However, the required use of box or point prompts is unsuitable for tumor segmentation. This is because the boundaries of tumors can be highly irregular and difficult to delineate in CT scans. This complexity necessitates extensive box or point labeling to obtain accurate masks, making the process labor-intensive.

To address the aforementioned issues, this work introduces an efficient language-guided adaptive cross-attention fusion framework that integrates adaptive modules specifically designed for tumors. Our model not only largely preserves the performance of the pre-trained model but also more effectively leverages the unique characteristics of data collected in local hospital settings. By incorporating these adaptive modules, our framework achieves an average DSC score of xxxx% on the MICCAI FLARE 2024 Challenge Task 1 dataset segmentation task.

## 2    Method

We make full of pretrained CLIP model and Univer model for our lesion segmentation.

### 2.1    Preprocessing

We studied the characteristics of the number of lesions within samples by analyzing the connected regions in the dataset and defined samples with single connected region lesions as single-lesion samples, while the rest were defined as multi-lesion samples. Among the 5,000 cases in train unlabeled, there are 2,397 single-lesion samples and 2,603 multi-lesion samples. There are significant differences between the various datasets. For example, the PETCT (whole body) dataset contains the most lesions, with a single case having 1,046 connected

regions; the coronacases dataset consists entirely of multi-lesion data; relatively, the MSD pancreas dataset has only two multi-lesion cases.

Additionally, we conducted further analysis on the Train unlabeled portion of the data. The AMOS dataset includes samples from 16 categories such as spleen, right kidney, left kidney, gallbladder, etc., while the RSNA2023 dataset mainly covers injuries and internal bleeding of organs such as the liver and spleen. In contrast, the Validation Public dataset includes the FLARE23Ts 2023 abdominal multi-organ segmentation dataset and the LNDb lung nodule (lung cancer) dataset. Due to the diversity in sample categories and the number of lesions in the dataset, training a relatively general model faces considerable challenges.

Therefore, We split all the labeled data to 15 category according to the different type of lesions. And we encoded the text description of each lesions into the clip encoder. Such as

1. "Novel Coronavirus Pneumonia",

2. "Kidney lesions and Bone lesions and Pulmonary nodules and Swollen lymph nodes",

3. "Kidney Tumor and Kidney Cyst",

4. "Lung nodules",

5. "abdominal trauma with visceral organ injury and internal bleeding, including liver, spleen, kidneys, and intestines",

6. "Adrenocortical carcinoma" ,

7. "mediastinal lymph-nodes and celiac lymph node",

8. "Non-small cell lung cancer and Pleural effusion",

9. "whole body cancer or tumor",

10. "colon Tumor",

11. "pancreas Tumor",

12. "Hepatic Vessel Tumor",

13. "lung tumor",

14. "liver tumor",

15. "Spleen tumor, Right kidney tumor, Left kidney tumor, Gallbladder tumor, Esophagus tumor, Liver tumor, Stomach tumor, Aorta tumor, Postcava tumor, Pancreas tumor, Right Adrenal Gland tumor, Left Adrenal Gland tumor, Duodenum tumor, Bladder tumor, and Prostate/Uterus tumor".

Then, we adjust the window width and level to meet the appropriate Hounsfield unit for tumors, ranging from -700 to 300. Then we use isotropic spacing [1.5,1.5,1.5] and uniformed intensity scale to reduce the domain gap among various datasets

## 2.2   Proposed Method

CLIP (Contrastive Language-Image Pre-training) is a pretraining method developed by OpenAI [34]. Built upon the methodology of contrastive pre-training [21], it jointly optimizes a vision encoder and a text encoder, where the vision encoder is based on either ResNet [12] or Vision Transformer(ViT) [6]. The language encoder is rooted in a transformer-based model like BERT [5], forcing the paired image-text information to be as close as possible to the joint

image-text latent space after encoding. We adopt the original CLIP model as our text embedding extractor. Trained on a vast collection of image-text pairs, CLIP learns visual representation through text supervision, known as prompt. We design a specialized prompt for our pulmonary vessel segmentation task, as seen in Table. 1.

### 2.3 Pretrained text encoder and vision model

**Text encoder:** We use the original pre-trained CLIP encoder $E_{text}$ with a specially designed medical prompt ( *i.e.* 'A computerized tomography of a category with cancers and lesions') to generate text embeddings $H_t \in \mathbb{R}^{K*D}$, where K represents the number of class, and D represents the length of the embedding. The pre-trained encoder consists of a 12-layer 512-wide transformer with eight attention heads. The 512-wide output of the transformer is used as text embedding. We observe that the selection of medical prompt templates is hand-crafted and worthy of experiments. Table 1 illustrates the effectiveness of four different prompt templates. The last template, specifically designed for our vascular-shaped data, demonstrates nearly a 0.1% improvement compared to other commonly used templates, indicating that adjusting the prompt benefits our model.

**Table 1.** Ablation studys of different prompts.

| Embedding | prompt | DSC(%) ↑ | NSD(%) ↑ |
|-----------|--------|----------|----------|
| CLIP v1 | A photo of a category | 16.72 | 11.89 |
| CLIP v2 | A computerized tomography of a category | 17.75 | 11.10 |
| CLIP v3 | A computerized tomography of a category with cancers and lesions | 17.77 | 12.94 |

**Vision model:** Medical Segmentation Decathlon [1] is a benchmark for many medical organ segmentation tasks. Specifically, Liu et al. [19] ranked first with an open-source pre-trained model [3] U-Net and Swin UNETR. Accounting for its strong ability to segment organs, the pre-trained model minimizes the time cost of training a model and inherits the weights that are suitable for organ segmentation. Therefore, we adopt a pre-trained U-Net model as the backbone for segmentation. Specifically, in our model, the 3D CT images $x_{img} \in \mathbb{R}^{H*W*L}$ are encoded into a feature map $V \in \mathbb{R}^{B*C*H*W*L}$ through the U-Net encoder $E_{img}$.

---

[3] https://github.com/ljwztc/CLIP-Driven-Universal-Model

$$H_v = E_{img}(x_{img}), \tag{1}$$
$$H_v^a = A_{img}(H_v), \tag{2}$$

To match the shape of $H_t$, we duplicate $H_v^a$ according to the class number K. We define:

$$\text{rep}(H, k) = concat[\underbrace{H, H, \ldots, H}_{k \text{ times}}], \tag{3}$$

then, we obtain the result $H_v^a \in \mathbb{R}^{B*K*D}$,

$$H_v^a = rep(A_{text}(H_v), K). \tag{4}$$

The alignment of image and text embedding $H_f$ uses a multi-layer perceptron (MLP) to generate parameters ($\theta_k$). Three sequential convolutional layers with $1 \times 1 \times 1$ kernels filling with ($\theta_k$) convert vision decoder output features $F$ into k predictions, where $P_k = Sigmoid((F * \theta_{k_1}) * \theta_{k_2}) * \theta_{k_3}), \theta_k = \{\theta_{k_1}, \theta_{k_2}, \theta_{k_3}\}$. $*$ represents convolution operation. For each class $k$, we get every foreground class $P_k \in \mathbb{R}^{1 \times H \times W \times L}$. After that, we merge k classes of prediction into one prediction $P$, shown in Fig 1. $P_k$ is supervised by label $Y_k$, where the overall loss is represented as:

$$\mathcal{L}_{sup} = \frac{1}{|B|} \sum_{i=1}^{|B|} \left[ \mathcal{L}_S(P_k, Y_k) \right], \tag{5}$$

where $\mathcal{L}_S = \frac{1}{2}\left[ \mathcal{L}_{Dice} + \mathcal{L}_{ce}\right]$; $\mathcal{L}_{Dice}$ and $\mathcal{L}_{ce}$ represent the Dice and cross-entropy losses, respectively.

Loss function: we use the summation between Dice loss and cross-entropy loss because compound loss functions have proven robust in various medical image segmentation tasks [22].

**Please introduce your strategies to reduce false positives on CT scans from healthy patients.** We dont́ use any strategies to reduce false positives on CT scans from healthy patients.

**Please introduce your strategies to deal with the partial labels.** We use the MLP to generate parameters to get 15 categories of lesion, results are combined as one output.

**Please introduce your strategies to use the unlabeled images.**

Unlabeled images and pseudo labels generated by the FLARE23 winning algorithm [39] were not used.

**Please introduce your strategies to improve inference speed and reduce resource consumption** We use monai framework to implement our method, Amount of overlap between scans along each spatial dimension are adjusted on our test cases ranged from 0 to 1; Device for the stitched output prediction of cpu or gpu is also adjusted. The best performance are bolded and used in our test time submission.

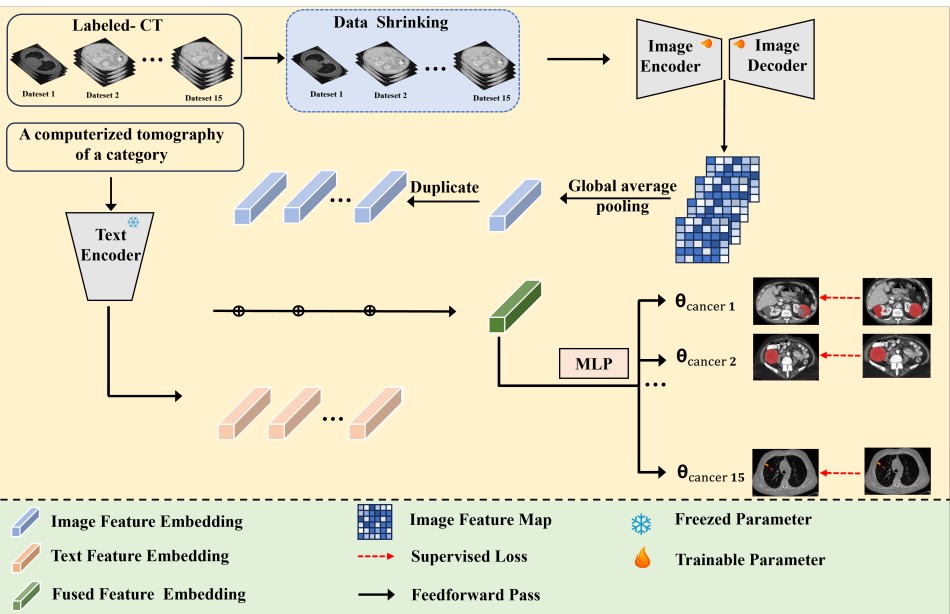

**Fig. 1.** Language-guided Vision Model architecture.

**Table 2.** Inference speed exploring table

| overlap | time | max_gpu |
|---------|------|---------|
| gpu | | |
| 0.1 | 47s | 9.8G |
| 0.2 | 41s | 9.8G |
| 0.3 | - | Out of Memory |
| 0.4 | 59s | 10.2G |
| 0.5 | 75s | 9.2G |
| cpu | | |
| 0.1 | 47s | 4.8G |
| **0.2** | **46s** | **4.8G** |
| 0.3 | 56s | 5.2G |
| 0.4 | 65s | 5.3G |
| 0.5 | 86s | 4.9G |

### 2.4   Post-processing

We do not use any special post-processing methods.

## 3   Experiments

### 3.1   Dataset and evaluation measures

The segmentation targets cover various lesions. The training dataset is curated from more than 50 medical centers under the license permission, including TCIA [4], LiTS [3], MSD [36], KiTS [13–15], autoPET [9, 10], TotalSegmentator [38], and AbdomenCT-1K [28], FLARE 2023 [27], DeepLesion [41], COVID-19-CT-Seg-Benchmark [26], COVID-19-20 [35], CHOS [17], LNDB [32], and LIDC [2]. The training set includes 4000 abdomen CT scans where 2200 CT scans with partial labels and 1800 CT scans without labels. The validation and testing sets include 100 and 400 CT scans, respectively, which cover various abdominal cancer types, such as liver cancer, kidney cancer, pancreas cancer, colon cancer, gastric cancer, and so on. The lesion annotation process used ITK-SNAP [43], nnU-Net [16], MedSAM [24], and Slicer Plugins [7, 25].

The evaluation metrics encompass two accuracy measures—Dice Similarity Coefficient (DSC) and Normalized Surface Dice (NSD)—alongside two efficiency measures—running time and area under the GPU memory-time curve. These metrics collectively contribute to the ranking computation. Furthermore, the running time and GPU memory consumption are considered within tolerances of 45 seconds and 6 GB, respectively.

### 3.2   Implementation details

**Environment settings** The development environments and requirements are presented in Table 3.

**Table 3.** Development environments and requirements.

| | |
|---|---|
| System | e.g., Ubuntu 18.04.5 LTS= |
| CPU | e.g., Intel(R) Core(TM) i9-7900X CPU@3.30GHz |
| RAM | 64GB; 2.67MT/s |
| GPU (number and type) | NVIDIA A100 40G |
| CUDA version | e.g., 12.0 |
| Programming language | e.g., Python 3.20 |
| Deep learning framework | e.g., torch 2.0, torchvision 0.2.2 |
| Specific dependencies | |
| Code | |

**Training protocols** We utilize a small set of labeled data, categorizing it into 15 classes to address partial labeling issues in lesion segmentation. To optimize the imaging parameters for tumor detection, we adjust the window width and level to a range of -700 to 300 Hounsfield units. Additionally, we apply isotropic spacing of [1.5, 1.5, 1.5] and standardize the intensity scale to minimize the domain gap across different datasets.

**Table 4.** Training protocols.

| | |
|---|---|
| Network initialization | pretrained model initialization |
| Batch size | 2 |
| Patch size | 96×96×96 |
| Total epochs | 300 |
| Optimizer | SGD |
| Initial learning rate (lr) | 8e-4 |
| Lr decay schedule | Cosine |
| Training time | 72.5 hours |
| Loss function | CE and DICE loss |
| Number of model parameters | 41.22M[4] |
| Number of flops | 59.32G[5] |
| $CO_2$eq | 1 Kg[6] |

## 4   Results and discussion

The main limitation of our algorithm lies in the significant differences in segmentation accuracy among different tumors, and the utilization of unlabelled data is relatively superficial. Developing new methods to improve the accuracy of tumor segmentation would be valuable. Additionally, since the tumor categories are not provided in the dataset, our method relies on statistical classification based on the available data, which is very rough, leading to issues such as low accuracy and poor robustness of the segmentation results.

In the future, we will continue to focus on segmenting whole-body tumors in CT scans. We will further investigate semi-supervised methods for pan-cancer CT scan segmentation. Specifically, we will concentrate on how to better utilize unlabelled data and improve the segmentation of small targets.

### 4.1   Quantitative results on validation set

In our approach, we initially employed the entire dataset for training, which included both labeled datasets and unlabeled datasets with pseudo-labels, totaling over 8,500 CT scan images. However, this approach did not yield satisfactory

**Table 5.** Quantitative evaluation results of our method.

| Method | FLARE23 | | LNDb | | Average | |
|---|---|---|---|---|---|---|
| | DSC(%) | NSD(%) | DSC(%) | NSD(%) | DSC(%) | NSD (%) |
| Small Model | 32.92 | 24.25 | 2.62 | 1.62 | 17.77 | 12.94 |
| Big Model | 18.37 | 17.29 | 6.04 | 5.31 | 12.21 | 11.3 |

results. We hypothesize that this may be due to the model's difficulty in effectively processing such a large volume of data for learning. To address this issue, we curated a smaller subset from the large dataset. Specifically, within the labeled dataset, we selected 120 CT samples from each category; if a category had fewer than 120 samples, all available samples were chosen. Considering that each dataset contained CT images from different categories, we utilized stratified proportional sampling to ensure the representativeness of the samples. Through this method, we ultimately constructed a labeled dataset consisting of 1,012 CT scan images.

Table 5 presents the quantitative results of our method when trained using the full dataset and small batches of data. The public validation set includes two subsets: FLARE23 and LNDb, corresponding to abdominal organ tumors and lung nodules, respectively. We calculated the DSC and NSD for both subsets.When trained on the full dataset, our model achieved an average DSC of only 12.21%. In contrast, when trained on small batches of data, the average DSC and average NSD were 17.77% and 12.94%, respectively. Despite using pseudo-labels from additional unlabelled data, the performance did not improve as expected, suggesting that the model may struggle with handling such large volumes of data.Further analysis shows that the DSC for the FLARE23 subset increased from 18.37% to 32.92%, while the DSC for the LNDb subset decreased from 6.04% to 2.62%. Based on statistical analysis of the datasets, we observed that abdominal tumor samples are more prevalent, whereas the LNDb dataset focuses on smaller lung nodules. These results indicate that our model performs well on abdominal tumors but exhibits lower segmentation accuracy on smaller targets.

**Table 6.** Quantitative evaluation of segmentation efficiency in terms of the running them and GPU memory consumption.

| Case ID | Image Size | Running Time (s) | Max GPU (MB) | Auc GPU Time (ms) |
|---|---|---|---|---|
| 0001 | (512, 512, 55) | 31.87 | 6153 | 63808 |
| 0051 | (512, 512, 100) | 18.54 | 6153 | 44506 |
| 0017 | (512, 512, 150) | 18.49 | 6169 | 44731 |
| 0019 | (512, 512, 215) | 20.06 | 7137 | 53396 |
| 0099 | (512, 512, 334) | 25.75 | 6225 | 78071 |
| 0063 | (512, 512, 448) | 19.28 | 6185 | 47336 |
| 0048 | (512, 512, 499) | 25.45 | 7121 | 65130 |
| 0029 | (512, 512, 554) | 22.47 | 6153 | 55571 |

## 4.2   Qualitative results on validation set

Fig. 2 presents two examples of successful segmentation results and two examples of poor segmentation outcomes, using our method trained on the entire dataset and on small batches. In Case #LNDb_0004, both approaches achieved accurate tumor segmentation. In Case #FLARE23Ts_0013, the small batch training method successfully segmented the tumor, whereas the method trained on the entire dataset failed to do so effectively. However, in Case #FLARE23Ts_0035 and Case #LNDb_0066, neither approach was able to accurately segment the tumor.

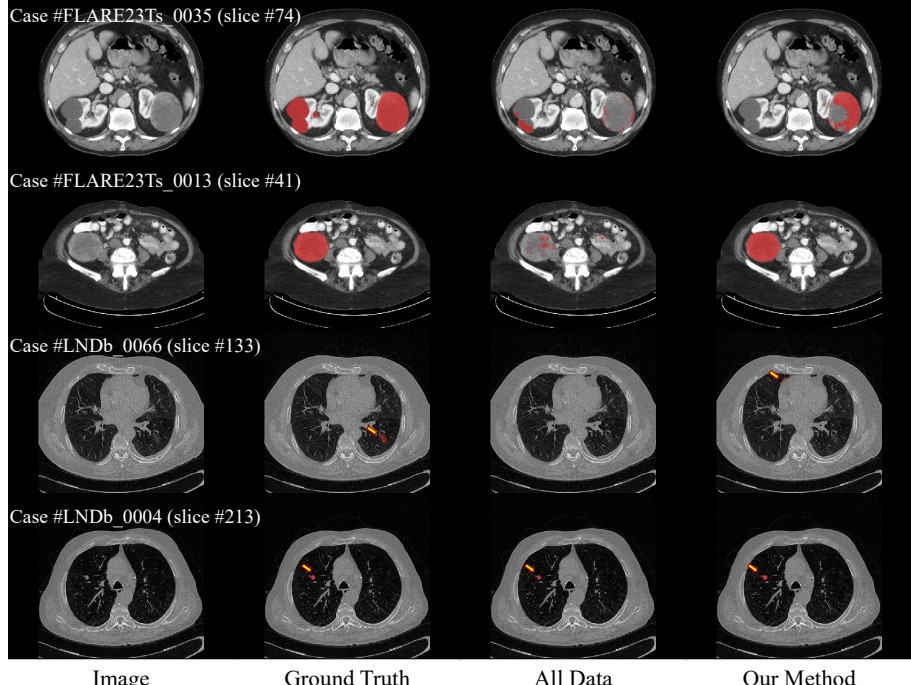

**Fig. 2.** Qualitative results of four examples provided by training our method on both the entire dataset and smaller batches are presented. Case #FLARE23Ts_0013 and Case #LNDb_0004 illustrate successful segmentation outcomes, while Case #FLARE23Ts_0035 and Case #LNDb_0066 demonstrate poor segmentation results. The arrows indicate the segmented regions.

'

## 4.3   Segmentation efficiency results on validation set

We report our segmentation efficiency methods in Table. 2

### 4.4   Results on final testing set

This is a placeholder. We will send you the testing results during MICCAI.

### 4.5   Limitation and future work

The main limitation of our algorithm lies in the significant differences in segmentation accuracy among different tumors, and the utilization of unlabelled data is relatively superficial. Developing new methods to improve the accuracy of tumor segmentation would be valuable. Additionally, since the tumor categories are not provided in the dataset, our method relies on statistical classification based on the available data, which is very rough, leading to issues such as low accuracy and poor robustness of the segmentation results.

In the future, we will continue to focus on segmenting whole-body tumors in CT scans. We will further investigate semi-supervised methods for pan-cancer CT scan segmentation. Specifically, we will concentrate on how to better utilize unlabelled data and improve the segmentation of small targets.

## 5   Conclusion

We make use of the clip univer model to segment lesions. Split the dataset into 15 categories and select a rather samll CT dataset get a more higher performance. We select overlap=0.2 and output device = "cpu" to accelerate our inference speed as well as lowing gpu comsumpution.

**Acknowledgements** The authors of this paper declare that the segmentation method they implemented for participation in the FLARE 2024 challenge has not used any pre-trained models nor additional datasets other than those provided by the organizers. The proposed solution is fully automatic without any manual intervention. We thank all data owners for making the CT scans publicly available and CodaLab [40] for hosting the challenge platform.

## Disclosure of Interests

Authors must declare any competing interests or specifically state that the authors have no competing interests in the paper as required by the publisher. If no COI, please direct say "The authors declare no competing interests.".

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

**Table 7.** Checklist Table. Please fill out this checklist table in the answer column.

| Requirements | Answer |
| --- | --- |
| A meaningful title | Yes/No |
| The number of authors ($\leq 6$) | Number |
| Author affiliations and ORCID | Yes/No |
| Corresponding author email is presented | Yes/No |
| Validation scores are presented in the abstract | Yes/No |
| Introduction includes at least three parts: background, related work, and motivation | Yes/No |
| A pipeline/network figure is provided | Figure number |
| Pre-processing | Page number |
| Strategies to use the partial label | Page number |
| Strategies to use the unlabeled images. | Page number |
| Strategies to improve model inference | Page number |
| Post-processing | Page number |
| The dataset and evaluation metric section are presented | Page number |
| Environment setting table is provided | Table number |
| Training protocol table is provided | Table number |
| Ablation study | Page number |
| Efficiency evaluation results are provided | Table number |
| Visualized segmentation example is provided | Figure number |
| Limitation and future work are presented | Yes/No |
| Reference format is consistent. | Yes/No |