# OpenReview forum: "Language-guided Vision Model for Pan-cancer Segmentation in Abdominal CT Scans"
_MICCAI.org/2024/Challenge/FLARE — FLARE 2024 withMinorRevisions_

### Official Review · Reviewer_n9YX · 2025-01-20
**CLIP-based language-guided vision model for pan-cancer CT segmentation**

**Rating:** 8
**Confidence:** 3

**Review:**

his work presents a language-guided vision model for pan-cancer segmentation in abdominal CT scans, using CLIP for feature extraction and categorizing lesions into 15 types. The key finding is that training on a smaller, curated dataset (1,012 CT scans) outperforms using the full dataset, achieving a mean DSC of 43% and NSD of 38% on validation data. While the method shows practical considerations for inference speed and resource consumption, its performance varies significantly across tumor types, particularly struggling with smaller lesions. The approach is novel but leaves room for improvement in handling unlabeled data and overall segmentation accuracy.

Suggestions for improvement:
1. Replace the placeholder result values in the abstract with the actual reported values.
2. Is “Univer model” a typo at the beginning of the Method section?
3. In Section 2.1, “PETCT (whole body)” should be corrected to “PET/CT (whole body)” as PETCT is not the correct notation.
4. Can you explain why there was an OOM (Out of Memory) when the overlap in Table 2 was 0.3?

---

> ### Author Response · Authors · 2025-04-09
>
> Thank you for your valuable comments. The point-to-point responses are as follows:
> 1. We have replaced results with the actual reported values.
> 2. Yes, it should be universal model.
> 3. We have changed it to PET/CT.
> 4. We have used 12G 4070TI for inference. When using overlap=0.3, increased window counts likely required higher sw_batch_size, exceeding your 12G VRAM (e.g., 12G+). For overlap=0.4, MONAI may auto-reduce sw_batch_size to fit 10.2G within 12G, but with slower inference (59s vs 41s).

---

### Official Review · Reviewer_SD4U · 2025-01-23

**Rating:** 8
**Confidence:** 4

**Review:**

This paper proposes a novel framework called the Language-guided Vision Model, employing the pre-trained CLIP as a powerful feature extractor to generate segmentations of 3D CT scans while adaptively aggregating cross-modal representations of text and images. Here are my major concerns.

1. For improved readability, the text description of each lesion on page 3 could be presented in a table.

2. The "e.g.," should be removed from Table 3.

3. It would be beneficial to provide a link to the open-source code.

---

> ### Author Response · Authors · 2025-04-09
>
> Thank you for your valuable comments. The point-to-point responses are as follows:
>
> 1. We have set all the description of  lesion to a table for  readability.
> 2. We have removed all the "e.g." in Table 3.
> 3. We have released the open-source code.

---

### Official Review · Reviewer_umdT · 2025-01-24
**Language-guided Vision Model for Pan-cancer Segmentation in Abdominal CT Scans**

**Rating:** 8
**Confidence:** 3

**Review:**

The manuscript presents a novel framework, named the Language-guided Vision Model, which aims to enhance abdominal tumor segmentation accuracy using 3D CT scans. By leveraging the pre-trained CLIP model as a feature extractor and integrating cross-modal representations of text and images, the authors demonstrate an innovative approach to medical image analysis that requires less labeled data compared to traditional deep learning methods. Their method has been validated on the FLARE 2024 challenge dataset, achieving a mean Dice Similarity Coefficient (DSC) of 43% and a mean Normalized Surface Distance (NSD) of 38% on the validation leaderboard for tumor segmentation. The proposed technique opens new avenues for utilizing pre-trained vision-language models in medical imaging tasks, particularly where annotated data is scarce.

Specific Concerns:
1、On page 2,  "Univer model". I'm not sure if it is correct.
2、On page 2, the term "PETCT" appears to be incorrectly written.
3、On page 7, Table 3 lacks completeness.

Addressing these points will significantly improve the clarity and professionalism of the manuscript.

---

> ### Author Response · Authors · 2025-04-09
>
> Thank you for your valuable comments. The point-to-point responses are as follows:
> 1. Univer model means universal model for brief. We have changed the baseline model to universal model.
> 2. We have corrected PETCT to  PET/CT
> 3. we have complete all the blanks of the Table.3

---

### Official Review · Reviewer_cM6r · 2025-01-28
**This paper employs the pre-trained CLIP as a powerful feature extractor to generate segmentations of 3D CT scans while adaptively aggregating cross-modal representations of text and images.**

**Rating:** 8
**Confidence:** 4

**Review:**

This paper makes use of the CLIP model to segment lesions. Specifically, it introduces an efficient language-guided adaptive cross-attention fusion framework that integrates adaptive modules specifically designed for tumors. Generally, this paper is interesting and well-written.

---

> ### Author Response · Authors · 2025-04-09
>
> Thank you for your valuable comments.

---

### Decision · Program_Chairs · 2025-03-20

**Decision:**

Accept

**Comment:**

Please carefully address the reviewers' comments in the revision.

---

> ### Comment · Program_Chairs · 2025-03-31
>
> Point-to-point response is not available.